# Biocontrol Activity of *Aureubasidium pullulans* and *Candida orthopsilosis* Isolated from *Tectona grandis* L. Phylloplane against *Aspergillus* sp. in Post-Harvested Citrus Fruit

**Dalia Sukmawati** [1,2,*], **Nurul Family** [1], **Iman Hidayat** [3], **R. Z. Sayyed** [4], **Elsayed A. Elsayed** [5,6], **Daniel Joe Dailin** [7,8], **Siti Zulaiha Hanapi** [7], **Mohammad A. Wadaan** [5] **and Hesham El Enshasy** [7,8,9,*]

1   Biology Department, Faculty of Mathematics and Natural Sciences, Universitas Negeri Jakarta, Jakarta 13220, Indonesia; familinurul@gmail.com
2   Universitas Negeri Culture Collection, Faculty of Mathematics and Natural Sciences, Universitas Negeri Jakarta, Jakarta 13220, Indonesia
3   Microbiology Division, Research Centre for Biology, Indonesian Institute of Sciences (LIPI), Bogor 16911, Indonesia; oxydothis@gmail.com
4   Department of Microbiology, PSGVP Mandal's, Arts, Science & Commerce College, Shahada 425409, India; sayyedrz@gmail.com
5   Bioproducts Research Chair, Zoology Department, College of Science, King Saud University, Riyadh 11451, Saudi Arabia; eaelsayed@ksu.edu.sa (E.A.E.); wadaan@ksu.edu.sa (M.A.W.)
6   Chemistry of Natural and Microbial Products Department, National Research Centre, Cairo 12622, Egypt
7   Institute of Bioproduct Development (IBD), Universiti Teknologi Malaysia (UTM), Skudai 81310, Malaysia; jddaniel@utm.my (D.J.D.); zulaiha@ibd.utm.my (S.Z.H.)
8   School of Chemical and Energy Engineering, Faculty of Engineering, Universiti Teknologi Malaysia, Skudai 81310, Malaysia
9   City of Scientific Research and Technology Applications (SRTA), New Burg Al Arab, Alexandria 21934, Egypt
*   Correspondence: Dalia-Sukmawati@unj.ac.id (D.S.); henshasy@ibd.utm.my (H.E.E.)

**Abstract:** This study aimed to isolate and identify moulds from rotten *Citrus sinensis* post-harvests and to investigate the activity of antagonist and biocontrol activity moulds that cause citrus fruit rotting. A total of 12 mould isolates were obtained. Following the pathogenicity test, two representative mould isolates were selected and identified based on the sequence analyses of internal transcribed spacer (ITS) regions of the rDNA. Methods used in this study include isolation of fungal postharvest diseases, pathogenicity assay, antagonism assay, growth curve analysis, in vitro biocontrol assay, and molecular phylogenetic analysis. Two isolates of fungal postharvest diseases were determined as the most destructive pathogens. The biocontrol assay showed that isolates of Y1 and Y10 were capable to reduce the growth of fungal isolates K6 and K9 and mitigate up to 100% of the damage of sweet citrus fruits after 7 days of incubation. The moulds were identified as K6 (*Aspergillus flavus* sensu lato) and K9 (*Aspergillus niger* sensu lato). Phylogenetic analysis showed that the Y10 yeast isolate was identified as *Candida orthopsilosis*, whereas the Y1 isolate had a close genetic relationship with *Aureobasidium pullulans* and possibly belongs to a new species. Further analysis is necessary to confirm this finding.

**Keywords:** *Aspergillus*; *Aureobasidium pullulans*; biocontrol; *Candida orthopsilosis*; post-harvest

## 1. Introduction

Damage that occurred to postharvest products results in reduced quality and nutrient content of fruit. This can occur during picking, storing, packing, and in circumstances that lack sanitation [1,2]. This can cause the colonization of moulds, causing damage to the fruit. Sangorrín et al. [3] stated that damage to post-harvest fruits is caused by pathogenic moulds, mainly *Alternaria*, *Aspergillus*, *Botrytis*, *Fusarium*, *Geotrichum*, *Gloeosporium*, *Mucor*, *Monilinia*, *Penicillium*, *Rhizopus*, and other genera. Citrus is a fruit that contains several

benefits such as antioxidants, vitamin C, hydroxycinnamic acids, and anthocyanins [4]. Both the fruit and the whole plant are vulnerable to pathogenic moulds [5].

Several destructive moulds infect citrus fruits during postharvest, namely *Aspergillus niger*, *A. flavus*, *A. fumigatus*, *A. terreus*, *Penicillium italicum* [6], *Penicillium digitatum* [7], *Penicillium verrucosum*, *Rhizopus arrhizus*, *Rhizopus stolonifer*, *Fusarium oxysporum*, *Fusarium solani*, *Alternaria alternata*, and *Mucor* sp. [8]. *Fusarium* sp. and *Aspergillus niger* causes rot on citrus fruits [9]. The mould of *P. digitatum* causes green weathered disease, which causes all parts of the fruit to be damaged [10]. In addition to causing rot, *P. digitatum* also produces mycotoxin patulin, which is harmful to human health [11,12].

The presence of destructive mould infections can reduce the quality and quantity of citrus fruits that are ready to be sold. One method of mould control is the utilization of synthetic fungicides. Common examples include thiabendazole, imazalil, and sodium o-phenylphenate [13]. However, extended usage can cause destructive moulds to resist them and is harmful to consumers. Besides, poor management of fungicides can cause the accumulation of fungicidal residues, which leads to environmental pollution [14,15].

Destructive fungi can be naturally controlled using biological agents such as yeast [12,16,17]. Antagonism in some types of yeast can be used to inhibit infection and the growth of several types of destructive moulds [18–21]. Epiphytic yeast on the leaf surface (phylloplane) is one of the yeast groups that have the potential to be antagonized [22]. Tropical plant species, such as the teak tree (*Tectona grandis*), can be overgrown by phylloplane yeast. Teak trees are widely used as medicine for headaches, skin diseases, bronchitis, dysentery, anti-inflammation, and anti-diabetes. The leaves, roots, flowers, and fruits contain saponins, polyphenols, and flavonoids. It should be noted that saponins in teak leaves have an antifungal function [23]. The objective of this study was to investigate the effectiveness of yeast obtained from phylloplane of *T. grandis* as biocontrol agents of the citrus pathogenic mould in vitro and in vivo.

## 2. Materials and Methods

### 2.1. Microorganisms and Fruit Materials

#### 2.1.1. Yeasts

A total of 22 yeast isolates from the collection of the Jakarta Culture Collection State University (UNJCC) derived from the isolation of phylloplane *T. grandis* from teak leaves were utilized. Yeast cultures from the *T. grandis* plant provided by the UNJCC collection were used for antagonistic and biocontrol tests on pathogenic moulds in citrus fruits.

#### 2.1.2. Moulds

The pathogenic mould was a result of isolation from rotted fruit based on the results of pathogenicity tests. It was grown on oatmeal agar (60 g of oatmeal, 10 g of sodium nitrate, 30 g of sucrose, and 12 g of agar per 1 L of distilled water) and incubated at 25 °C for 5 days. Pathogen conidia were collected and suspended in sterile distilled water containing 0.05% ($v/v$) Tween 80 and the suspension was adjusted to $10^4$ cfu/mL, with a hemocytometer.

#### 2.1.3. Fruit

*Citrus sinensis* fruits obtained from Jatinegara Market in DKI Jakarta (Indonesia) were used. Fruits without physical defection or infection were sorted based on size. Selected fruits were surface-disinfected with distilled water (5 min), NaOCl 0.5% (1 min), and alcohol 70% (1 min). The fruits were washed with sterile distilled water and air-dried before the pathogenicity test. They were divided into two; those that were wounded with a sterile nail (2 mm$^3$) on the equator (one wound) and those that were wounded on the other side of the fruit.

### 2.2. Isolation Mould from Rotted Citrus

A total of 15 infected citrus fruits were obtained through the purposive sampling method from Jatinegara market, East Jakarta, Indonesia. According to Semangun [9], there

are several infected fruit characteristics, namely, soft flesh, brownish fruit skin, blackish circles, and moulded mycelium. The infected fruits were brought immediately to the laboratory using a cool box. The moulds were then isolated by administering a dental needle containing potato dextrose agar (PDA) and then sampled in two snippets. The snippets containing pathogenic mould mycelium were incubated on PDA at 28 °C for 7 days. Mould isolates that were grown were then purified using the hyphal tips method.

### 2.3. Pathogenicity Test

Pathogenicity testing was carried out based on the Koch Postulate method with mould inoculation using the wound method. The mould isolates being tested were results from isolation. The sterilized fruit was cut on two sides of the surface using a syringe driver with a ± 1 cm stroke length. After that, mould spores were applied to the wound using a brush. Following this, the fruits were placed in a plastic tub with wet cotton on the four corners to maintain moisture and then incubated at 28 °C. Observations on the percentage of occurrence and severity of the disease were made 7 days after inoculation. Scoring was carried out according to Embaby et al. [24] using the following disease score scale: 0 = no symptoms of fruit rot; 1 = rotten fruit with a diameter of 0.5 cm without sporulation; 2 = rotten fruit with a diameter of 0.5–1 cm with sporulation; 3 = rotten fruit with a diameter of 1.1–2.5 cm with sporulation; 4 = rotten fruit with a diameter of 2.6–4 cm with sporulation; and 5 = fully rotten fruit containing mycelium.

### 2.4. Yeast Cell Growth Curve Development for Biocontrol Test

The yeast cell growth curve was obtained by counting the number of yeast cells using a hemocytometer. The media was a modification of the solution used by Sarlin and Philip [25].

### 2.5. Screening of the Yeast Isolates for Antifungal Activity In Vitro

This assay was conducted utilizing the methodology described by Mahadtanapuk et al. [26]. The in vitro yeast antagonism test was conducted using the modified dual-culture method. Yeast and destructive moulds were inoculated on the MEA inside a petri dish. The yeast suspension was inoculated along the lines using a micropipette. The culture of moulds was taken using a modified dental needle. Moulds were inoculated parallel to the yeast with a distance of 2.5 cm on the MEA. The medium was then incubated at 28 °C for 5–6 days. The control was an MEA medium inoculated with hyphae/mould spores without inoculation of yeast suspension and an MEA medium inoculated without inoculation of hyphae/mould spores with the suspension of yeast colonies. Testing was conducted with three repetitions of the treatment.

### 2.6. In Vivo Antagonistic Activity Assays Fruits

In vivo testing of antagonism was carried out following a modified method of Sperandio [12]. Eight treatments were carried out, namely: (1) surface sterilization; (2) tap water washing; (3) no surface sterilization or washing (as is); (4) soaked with yeast antagonists; (5) inoculated with mould; (6) soaked with antagonist yeast and inoculated with mould; (7) soaked in distance M-45; and (8) soaked in distance M-45 and inoculated with mould. Each treatment was carried out five times with 10 oranges each to ensure the reproducibility of the results.

All fruits that were provided treatment were incubated in plastic boxes with each corner cotton-padded and slightly moistened beforehand. Incubation was carried out for 7 days at 28 °C. Observations were conducted every two days, with the parameters observed being disease incidence and disease severity. Scoring was based on Embaby et al. [24] using the following disease score scale: 0 = no symptoms of fruit rot; 1 = rotten fruit with a diameter of 0.5 cm without sporulation; 2 = rotten fruit with a diameter of 0.5–1 cm with sporulation; 3 = rotten fruit with a diameter of 1.1–2.5 cm with sporulation;

4 = rotten fruit with a diameter of 2.6–4 cm with sporulation; and 5 = fully rotten fruit containing mycelium.

### 2.7. Identification of Yeast Isolates by Amplification of the ITS Regions of the rDNA

This assay was conducted using the methodology described by Sukmawati et al. [20]. The isolation of yeast from phylloplane *T. grandis* was conducted in a YMA medium at 28 °C for 24 h, then the DNA was extracted using a Gneaid Genomic DNA Purification Kit. The amplification of the internal transcribed spacer (ITS) region of the rRNA gene, including the 5.8S gene, was performed by PCR using the primers: forward primer ITS5 (GGAAGTAAAAGTCGTAACAAGG) and reverse primer ITS4 (GGAAGTAAAAGTCG-TAACAAGG). The PCR reaction used the Mastermix Go Taq Green (Promega) for moulds and the KAPA3G Mastermix for yeast with a final volume of 25 μL. The composition for yeast consisted of 7.5 μL nuclease-free water (NFW); 1.5 μL MgCl$_2$; 0.5 μL dNTP; 12.5 μL buffer 3G; 0.5 μL DMSO; 0.5 μL of each primer (ITS5 dan ITS4); 0.5 μL *Taq polymerase*; and 2 colonies (1 μL) of DNA template. The composition for moulds consisted of 10 μL nuclease-free water (NFW); 12.5 μL Go Taq Green Mastermix (Promega); 0.5 μL of each primer (ITS5 dan ITS4); 0.5 μL DMSO; and 1 μL DNA template.

PCR conditions (Thermo Scientific™ Arktik™ Thermal Cycler, Vantaa, Finland) for moulds and yeasts were as follows: 95 °C for 3 min, followed by 35 cycles for denaturation at 95 °C for 30 s, annealing at 55 °C for 30 s, and elongation at 72 °C for 1 min. The final elongation was conducted at 72 °C for 10 min and soaking at 4 °C. PCR was sequenced to the sequencing service (1st BASE, Selangor, Malaysia). The results were analysed with BLAST at http://www.ncbi.nlm.nih.gov/, accessed on 22 April 2020, and then a phylogeny tree was created using the MEGA 7 program. The sequences obtained were aligned and compared with the National Center of Biotechnology Information (NCBI) database by the website using the Basic Local Alignment Search Tool (BLAST) [27].

## 3. Results

### 3.1. Isolation of Moulds

Ten Citrus fruit samples were obtained from the Jatinegara traditional market, East Jakarta. Citrus fruits used for mould isolation had a wrinkled fruit skin and a soft fruit texture and contained mycelium and spores in the fruit (Figure 1). According to Akinro [28], rotten citrus is characterized by a change in the colour of the fruit to a brownish, soft texture of the fruit and the production of gas. Twelve mould isolates were obtained as a result of the isolation of mould from rotten citrus. The mould isolates obtained showed sporulated black colonies (four isolates), sporulated green granules (four isolates), sporulated green velvet (two isolates), and greyish mycelium (two isolates) (Figure 2).

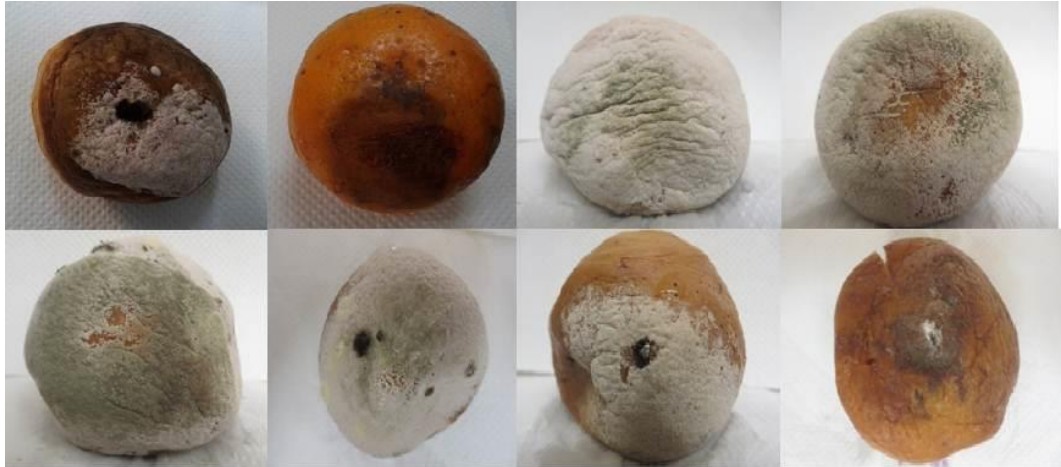

**Figure 1.** Different citrus fruits were used for the isolation of the pathogenic moulds.

**A**. Colony with black sporulation.

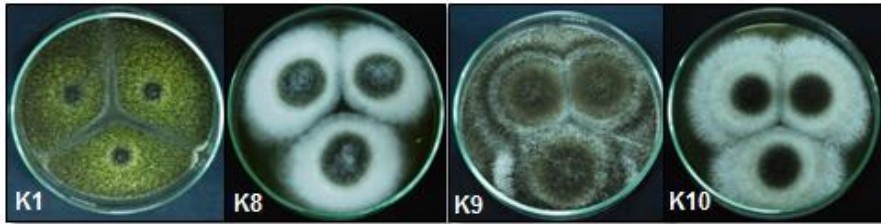

**B**. Colony with green sporulation.

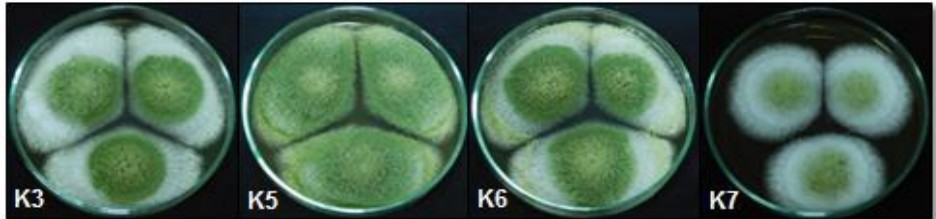

**C**. Colony with green velvety.        **D**. Colony with grey mycelium.

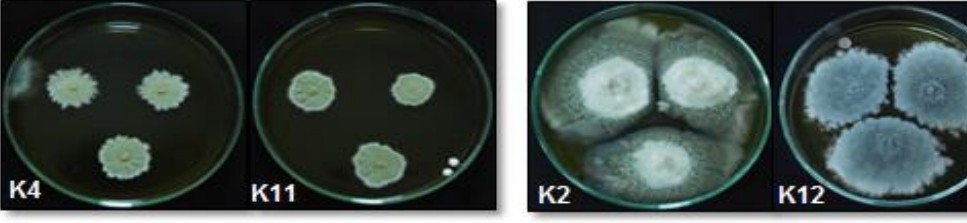

**Figure 2.** Mould isolates obtained from *Citrus*. Colonies on MEA after 7 days incubation at 28 °C.

### 3.2. Pathogenicity Test of Destructive Moulds with the Postulate Koch Method

The pathogenicity test utilized two inoculation methods, namely, wounding and topping. The results showed that inoculation with the wound method was more effective in producing decay symptoms. This test was conducted on eight isolates of isolated moulds (K1, K3, K5, K6, K7, K8, K9, and K10). Pathogenic mould parameters are shown by the percentage of disease and disease severity. Observations were made for 7 days of incubation at 28 °C (Figure 3).

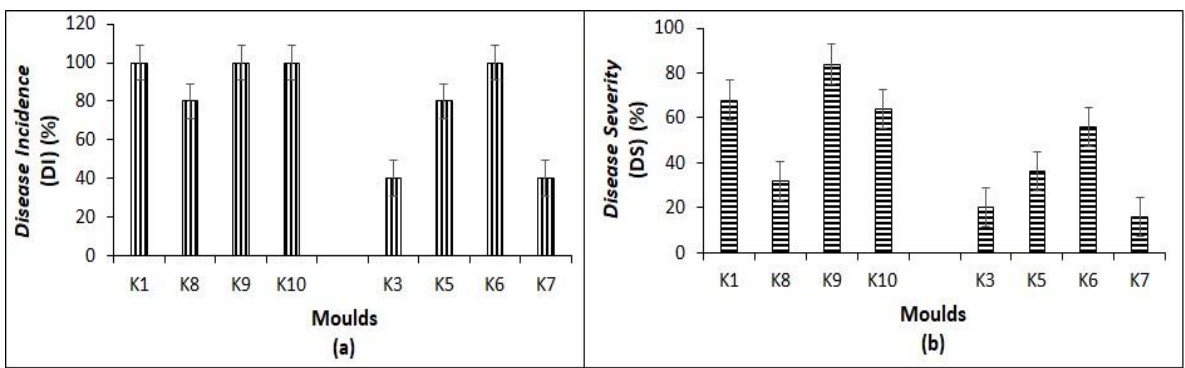

**Figure 3.** Results of pathogenicity test of mould isolate from rotten citrus with incubation of 7 days at 28 °C. K1-K10 = black sporulated mould isolates; K3-K7 = green spotted mould isolates. (**a**) Disease incidence (DI); (**b**) disease severity (DS). Error bars indicate standard deviations of experimental data (*n* = 5).

The results of this test showed that the highest fruit damage occurred in fruits inoculated with K9 (black sporulation) and K6 mould isolates (green sporulation) (Figure 3). The

highest disease incidence percentage was produced by black-spotted K9 (100%) and green spore K6 (84%), whereas the disease severity of K9 isolates (84%) and K6 isolates (56%) were superior.

Compared to seven other mould isolates. The pathogenicity of the two isolates was shown by the occurrence of decay in citrus fruits and a faster mould spore growth than other isolates (Figure 4).

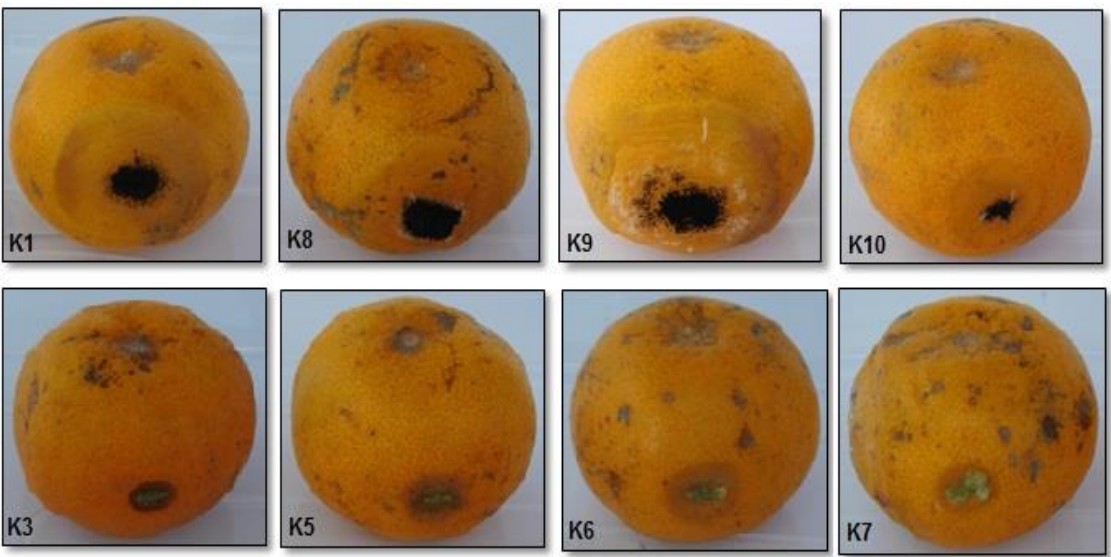

**Figure 4.** *Citrus* fruit pathogenic assay using the wounding method; 7 days of incubation at 28 °C; K1; K8; K9; K10 = black sporulation; K3; K5; K6; K7 = green sporulation.

### 3.3. Antagonism Test of Yeast from Teak Leaves against Destructive Moulds in Postharvest Citrus Utilizing the Dual Culture Method

Antagonism testing was carried out on 22 yeast isolates from teak leaves against two citrus destroyers (K6 and K9), utilizing a dual culture method based on Mahadtanapuk et al. [26]. The parameter observed in this test was the presence of inhibitory zones between mould colonies and yeast cells that were grown together on MEA. The results showed that from 22 yeast isolates, only two isolates (Y1 and Y10) had inhibitory zones against two destructive moulds, whereas the others did not. Y1 isolates had inhibitory zones for K6 (2.3 cm) and K9 (2.33 cm) mould isolates, whereas Y10 isolates had inhibition zones for K6 (1.5 cm) and K9 (1.6 cm) moulds (Figure 5).

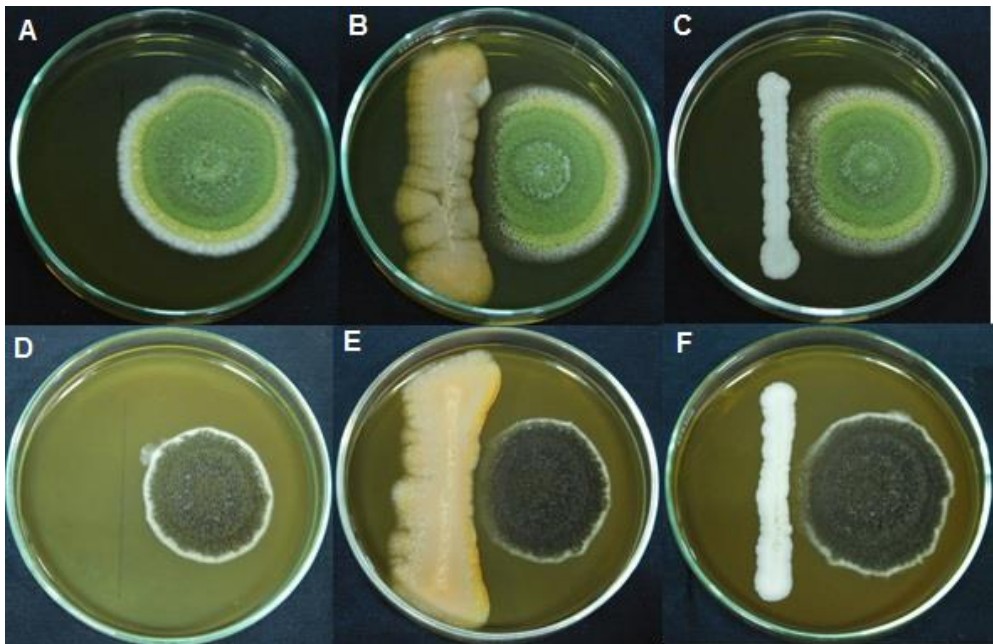

**Figure 5.** Antagonism test aged for 6 days of incubation on MEA at a 28 °C. (**A**) = K6 fungal control; (**B**) = yeast Y1 with fungal K6; (**C**) = yeast Y10 with fungal K6; (**D**) = K9 fungal control; (**E**) = Yeast Y1 with fungal K9; (**F**) =Yeast Y10 with fungal K9.

The results of the two-way ANOVA analysis (data not shown) indicate that there were differences in the width of the inhibitory zone between Y1 and Y10 yeast isolates against the test mould, with a sig. value of $0.00 < α$ (0.05). Based on the Duncan Multiple Range Test (DMRT) at the 5% level, it was observed that yeast and Y10 isolates had significant differences (1.5467 [b] ± 0.083) in the width of the mycelium inhibition zone of K6 and K9 mould isolates. The Y1 yeast isolate had the highest potential to inhibit the growth of the mould mycelium. Its inhibitory zone had an average width of 2.3167 [a] ± 0.083.

*3.4. Creation of Yeast Cell Growth Curve for Biocontrol Test*

The growth curves of Y1 and Y10 yeast isolates describe the phase of yeast cell growth and were divided into four phases for 51 h of observation: lag phase (adaptation), log phase (exponential), stationary phase (balanced), and death phase (decrease). The final log phase of the two isolates occurred at the 33rd hour, with a total of Y1 ($1.3 × 10^7$ cfu/mL) and Y10 cells ($4.55 × 107$ cfu/mL) (Figure 6). The final log phase based on this growth curve was used as a reference for the duration of yeast incubation in the biocontrol test. Based on the growth curve, it is known that different types of yeast have different growth characteristics. The Y1 isolate experienced all phases of growth for 51 h of incubation, broken down into phase lag (hours to 0–3), log (hours 3–33), stationary (hours 33–45), and death (hours 45–51), whereas Y10 isolates only experienced the log phase (hours to 0–33) and the stationary phase (hours 33–51).

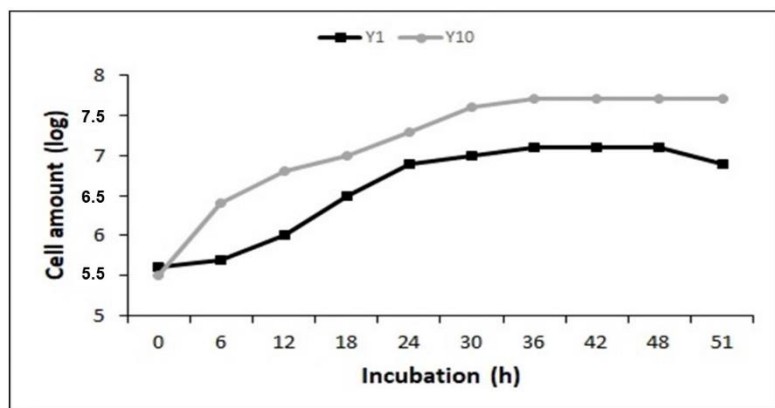

**Figure 6.** Y1 and Y10 yeast isolate growth curves incubated in NYDB for 51 h at 28 °C.

*3.5. Biocontrol Test of Yeast from Teak Leaves against Destructive Moulds in Postharvest Citrus*

Y1 and Y10 yeast isolates showed potential as biocontrol agents for K6 and K9 mould isolates, each showing different abilities from one another. Y1 isolates can reduce citrus fruit rot caused by the growth of K6 (50%) and K9 mould (40%), whereas Y10 isolates showed different percentage values of K6 (60%) and K9 mould (25%). The biocontrol ability of these isolates was significantly better than the dithane M-45 synthetic fungicide in inhibiting the growth of K6 and K9 moulds, thus reducing citrus fruit decay. The application of dithane M-45 0.3% fungicide cannot reduce the decay of citrus fruits, shown by the 100% percent of both K6 and K9 moulds in rotten citrus receiving this treatment.

The testing of antagonist yeast biocontrol on moulds showed a varying percentage of citrus rot. Y1 and Y10 yeast isolates can reduce the growth of K6 and K9 moulds, which in turn reduces rot by 100%. The wounding method was applied to the fruit before being soaked in the yeast suspension (Figure 7). The ability of Y1 and Y10 isolates to inhibit the growth of K6 and K9 moulds was significantly better compared to the dithane M-45 synthetic fungicide, which only reduced the growth of K9 mould isolates by 75% (data not shown).

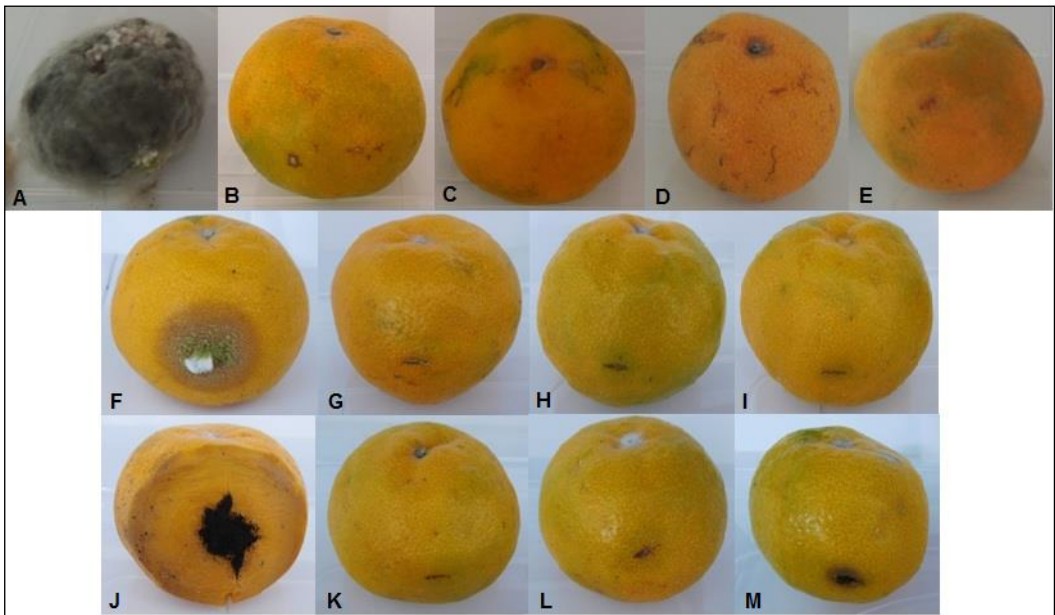

**Figure 7.** Biocontrol assay of yeast phylloplane derived from *T. grandis* against citrus moulds. Citrus was soaked in yeast/dithane following a 2 mm wounding. (**A**) = no treatment, (**B**) = washed with water, (**C**) = sterilized, (**D**) = Y1 yeast suspensions isolated, (**E**) = Y10 yeast suspensions isolated, (**F**) = K6 fungal control, (**G**) = Y1 yeast suspensions isolated + K6, (**H**) = Y10 yeast suspensions isolated + K6, (**I**) = dithane 0.3 % + K6, (**J**) = K9 fungal control, (**K**) = Y1 yeast suspensions isolated + K9, (**L**) = Y10 yeast suspensions isolated + K9, (**M**) = dithane 0.3 % + K9. Incubation at 28 °C of 7 days.

The application of yeasts Y1 and Y10 suspension before wounding showed varying results (Figure 7). The Y1 yeast isolate reduced the growth of K6 mould (50%) and K9 mould (40%), whereas Y10 reduced the growth of K6 mould (60%) and K9 mould (25%) by a different amount. The ability to inhibit the growth of K6 and K9 moulds of Y1 and Y10 yeast isolates was significantly better than the dithane M-45 synthetic fungicide, which, in turn, reduces the decay of citrus fruits. The application of dithane M-45 0.3% fungicide cannot reduce the decay of citrus fruits because of K6 and K9 moulds. This was shown by the 100% rotten citrus receiving this treatment of 100%, both on K6 and on K9 moulds (data not shown).

*3.6. Molecular (ITS Regions of rDNA) and Conventional (Morphological Characteristics of Colonies and Cells) Identification of Destructive Moulds and Potential Yeast*

The results of sequence analysis of the ITS rDNA region based on the phylogenetic tree (Figure 8) showed that the K6 mould isolate was identified as *Aspergillus flavus* sensu lato, which meant that it is part of a large *A. flavus* group. It is in one monophyletic clade with mould sequences consisting of *A. flavus* ATCC 16883, *A. minisclerotigenes* strain CBS 117635, *A. parvisclerotigenus* strain CBS 121.62, *A. fasciculatus* strain CBS 110.55, and *A. oryzae* isolates NRRL 447 with a bootstrap value of 87%. This shows that the six sequences have identical ITS nucleotide base sequences.

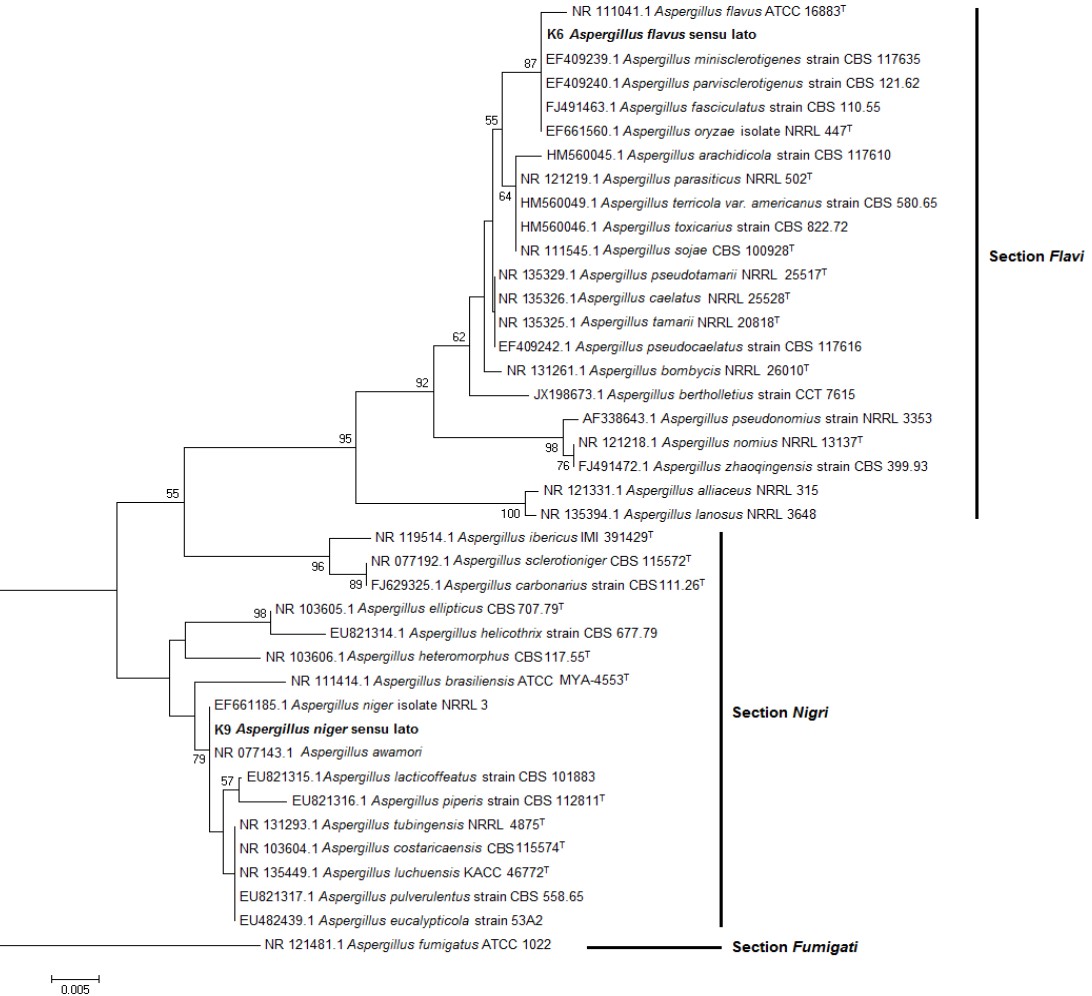

**Figure 8.** Phylogenetic tree of isolated postharvest mould disease based on the ITS region rDNA sequence. Bootstrap support values for 1000 replicates by the neighbour-joining method and MEGA 7 analysis. *A. fumigatus* ATCC 1022 in an outgroup.

The K9 mould isolate was identified as *A. niger* sensu lato, which meant that it is part of a large *A. niger* group. It is in one monophyletic clade consisting of *A. niger* NRRL 3 isolates; *A. awamori*. The three mould sequences have identical base sequences downloaded from NCBI (79% bootstrap).

The identification of molecular sequences of ITS rDNA regions was carried out on potential biocontrol yeast after postharvest. Y1 isolates were in the paraphyletic clade with *Aureobasidium pullulans, Kabatiella line*, and *A. namibiae*. The Y1 isolate sequence with the *A. namibiae* sequence downloaded from NCBI had a bootstrap value of 56%. This shows that the Y1 isolate had a homology level that is both not near and not far from *A. namibiae* (Figure 9). The Y10 isolate was identified as *Candida orthopsilosis* because it was in a monophyletic clade with a sequence of *C. orthopsilosis* with a 99% bootstrap value.

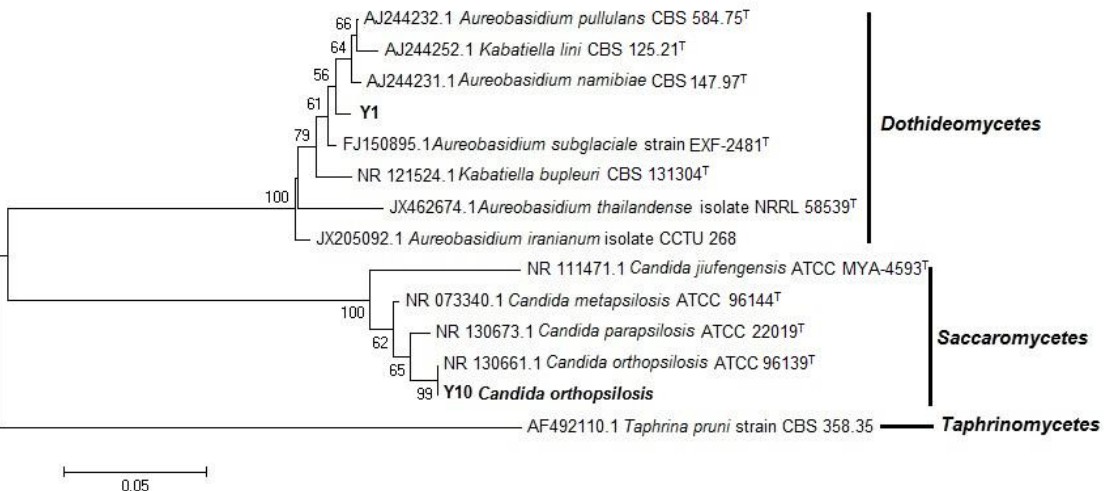

**Figure 9.** Phylogenetic tree of yeast phylloplane derived from isolated *T. grandis* based on the sequence of the ITS region rDNA. Bootstrap support values for 1000 replicates obtained by the neighbour-joining method and MEGA 7 analysis. *Taphrina pruni* CBS 358.35 in an outgroup.

## 4. Discussion

Citrus is a fruit commodity that is in great demand because of its high vitamin C content. However, the quality may be decreased because of the growth of destructive moulds in post-harvest. This study showed that damaged citrus fruits are rotten, wet, brownish in colour with a soft texture, and overgrown by mycelium mould. Postharvest fruits contaminated and damaged by mould are characterized by their soft texture, wrinkled skin, and the presence of mycelium and mould spores on the surface [9,28]. The *Aspergillus niger* causes *Aspergillus* rot, also called black mould rot [29]. This disease is spread through many spores found in the air and through contact between healthy and sick fruits. The initial symptom is the appearance of small, round, wet spots, which then enlarge and turn brown. Following this, white mycelium fungi appear. This can occur during the post-harvest process, in which the damage can be caused by many factors.

Damage to citrus fruits can occur during the handling, transportation, and storage of fruit [8,17,20]. The results of isolation in this study show that destructive moulds in citrus postharvest are characterized by their macroscopic size, black sporulation (four isolates), green sporulation granules (four isolates), and green sporulation (four isolates). Their morphological characteristics have similarities to *Aspergillus* and *Penicillium* genus fungi. Previous studies reported that these destructive moulds were a part of the *Aspergillus* and *Penicillium* genera [9,10,30].

Following this, the pathogenicity of eight isolates from moulds was tested. These isolates consisted of four green sporulated isolates (K3, K5, K6, and K7) and four black sporulated isolates (K1, K8, K9, and K10). The results of this test showed that the highest

damage occurred in fruit inoculated in K6 (green sporulation) and K9 (black sporulation) mould isolates. The K9 mould had a soft, smelly fruit texture, a brownish colour, and could grow black spores on the surface of the fruit. The testing of pathogenicity in citrus fruits conducted by Embaby et al. [24] showed that moulds with the highest cause of disease occurrence are characterized by their black to greyish spores and are identified as *Alternaria citri* and *Botryodia theobromae*. Oviasogie et al. [31] conducted a pathogenicity test on Nigerian citrus fruit and found symptoms of fruit rot, which are gas production, a soft texture, and a wrinkled and wet fruit surface.

The results of the sequence analysis of the ITS rDNA region with the phylogenetic tree (Figure 9) are additional proof, showing that the Y1 isolate is a new species. This is because the isolate Y1 is not in the monophyletic clade with its closest species sequence, but is actually in the paraphyletic clade with *Aureobasidium pullulans*, *Kabatiella line*, and *A.namibiae.* The Y1 isolate sequence compared with the *A.namibiae* sequence downloaded from NCBI had a bootstrap value of 56%, which shows that the Y1 isolate has a level of homology that is not near but also not far away from *A.namibiae.* The Y10 isolate was identified as *Candida orthopsilosis* because of its presence in a monophyletic clade with a sequence of *C. orthopsilosis* with a 99% bootstrap value.

It is known that *A.namibiae* is a variant of *A.pullulans*. According to Gostincar [32], *A.pullulans* has several variants that are analysed based on multilocus, namely: internal transcribed spacer rDNA, 28S rDNA, translation elongation factor-1$\alpha$ (EF1$\alpha$), $\beta$-tubulin, and elongase (ELO). The variants consist of *A. pullulans* var *pullulans*, *A. pullulans* var *melanogenum*, *A. pullulans* var *subglaciale*, *A. pullulans* var *namibiae*, and *A. pullulans* var *aubasidani.*

There have been no reports of *A.namibiae* as a potential biocontrol agent for destructive moulds. Meanwhile, *A.pullulans* are antagonistic yeasts that are used as biocontrol agents for destructive moulds on postharvest fruits and vegetables [32,33] and have been used as commercial products originating from Germany with the name Boni Protect [34]. The antagonism properties of *A. pullulans* are utilized as biocontrol agents because of their antifungal active compounds, namely, aureobasidin A [32]. Besides, it also produces several enzymes, namely amylase, protease, esterase, pectinase, xylanase, and mannase [35].

Similar to *Candida orthopsilosis*, there have been no reports of isolates derived from leaf surfaces (phylloplane) that are useful as biocontrol agents in postharvest fruits. However, some researchers have isolated and identified *C. orthopsilosis* yeast. Sukmawati et al. [20] have isolated and identified yeast phylloplane from *Broussonetia papyrifera* leaves in Java, Indonesia. Identification was carried out molecularly in the ITS area of rDNA, in which one of the yeasts identified was *Candida orthopsilosis*. Limtong [36] isolated and identified phylloplane yeast from *Saccharum officinarum* leaves in Thailand. Identification was carried out molecularly in the LSU rRNA (D1/D2) area, in which one of the yeasts identified was *C. orthopsilosis.*

The results of this study identified phylloplane derived from *Aureubasidium pullulans* (UNJCC Y1) and *Candida orthopsilosis* (UNJCC Y10) as antagonistic agents for *Aspergillus flavus* (UNJCC K6) and *Aspergillus niger* (UNJCC K9). *A. pullulans* yeast (UNJCC Y1) is more potent in inhibiting mycelium growth compared to *C. orthopsilosis* yeast isolate. This can be seen by the reduction of the diameter of the colony in the part facing the yeast cell. As shown by Sperandio et al. [12] and Moyano et al. [37], there was a growth reduction in moulds facing the potent yeast where it was able to reduce mycelium and form a sterile zone. The findings of this study converge with those of other authors, in which some of them used yeast as an antagonist agent [38,39].

The inhibiting mechanism can occur because of nutrient competition, antifungal compounds, and lytic enzymes such as glucanase, chitinase, and proteases [33,37]. The presence of lytic enzymes can cause the degradation of protein components that comprise mould cell walls, so that there is an inhibition of cell wall growth in mould mycelium. The enzymes produced by yeast cells can be chitinase enzymes and hydrolytic enzyme B-1,3-glucanase [40].

The inhibiting mechanism can occur because of competition from nutrients, antifungal compounds, and lytic enzymes such as glucanase, chitinase, and proteases. [37]. Growth activities of mould colonies may be disrupted because of a lack of nutrients and space to grow [33]. The presence of lytic enzymes can cause the degradation of protein components that comprise mould cell walls, which causes an inhibition of cell wall growth in the mycelium mould. The enzymes produced by yeast cells consist of chitinase and hydrolytic B-1,3-glucanase enzymes [40].

Biocontrol testing with a yeast cell density of $10^7$ cells/mL was able to inhibit the growth of *A. flavus* (UNJCC K6) and *A. niger* moulds (UNJCC K10). Yeast has a high growth rate in both room and low temperatures and acts as a biocontrol agent in citrus fruits [38,40]. Yeast phylloplane can be found in the extract of leaf surface and fruits and can survive in extreme conditions [20,41]. According to Li et al. [42], yeast isolates that are capable of colonizing fruit are important in the success of the biocontrol process. Moreover, Sperandio et al. [12] stated that space and nutrient competition are the main interactions of yeast isolates as biocontrol agents for postharvest disease. Another mechanism that might occur in the yeast inhibition process is nutrient competition. In low pH conditions, $Fe^{2+}$ in fruits will be oxidized to $Fe^{3+}$ to form $Fe(OH)_3$ complex compounds, which cause depletion (iron deficiency). Yeast has a strategy to overcome this iron deficiency by activating the Fe extraction system, mobilizing Fe storage from intracellular tissue, and utilizing metabolic adaptation to counteract the lack of Fe [43]. The ability of yeast to produce siderophore can bind iron compounds to the environment instead of the destructive moulds, which cause inhibition of mould growth [44]. Several studies have shown the capabilities of yeast as mould biocontrol agents in postharvest citrus fruits. Sperandio et al. [12] reported that the *A. pullulans* yeast was able to reduce damage to citrus fruits in Brazil caused by *P. digitatum* moulds. Ferraz et al. [39] reported that *A. pullulans, C. azyma*, and *R. minuta* were known to show biocontrol abilities against *Geotrichum citriaurantii* moulds in postharvest citrus fruits. It has also been reported that the yeast *Metschnikowia citriensis* exhibit protection activity against *Geotrichum citriaurantii* through the production of a direct antagonistic mechanism [45]. The postharvest protection of mandarins (*Citrus reticulata*) have also been reported through applying *Yarawia lipolytica*, which exhibit biocontrol activity against green and blue moulds [16]. A recent study also reported that the yeast *Clavispora lusitaniae* not only exhibits strong protection of citrus fruit against the postharvest infection by *P. digitatum* but also has the capacity to degrade patulin [11].

Apart from being able to survive extreme conditions, *A. pullulans* and *C. orthopsilisis* yeast can significantly reduce the decay of citrus fruit compared to the dithane M-45 fungicide 0.3%. The use of dithane M-45 is not recommended because it can cause irritation to the skin, eyes, and respiratory tract and can pollute the environment [46]. Therefore, this test shows that the use of yeast is superior to dithane M-45. Ultimately, it is possible to reduce the use of synthetic fungicides. The results of this study can provide information about yeast derived from teak leaves, which has the ability as a biocontrol agent that inhibits the growth of destructive moulds in postharvest citrus fruits. This research is expected to be able to become the basis of further research to obtain a biocontrol agent formulation that destroys postharvest citrus fruit.

## 5. Conclusions

Y1 and Y10 yeasts are excellent candidates for the formulation of alternatives to conventional fungicides for postharvest fungal disease control of green and blue moulds present in citrus postharvest. Based on these data, yeasts could be used for post-harvest preservation as a natural and safe alternative compared to the currently used fungicides, which have a high negative impact on environmental and human health.

**Author Contributions:** D.S.: conceptualization, writing—original draft, supervision, editing; N.F.: investigation, writing—original draft; I.H., data analysis, writing—original draft; R.Z.S.: formal analysis writing—review, editing; E.A.E.: methodology, resources, proofreading; D.J.D., formatting, review, proofreading; S.Z.H.: writing—review and editing, M.A.W.: resources, proofreading; H.E.E.:

facilitation, editing, review, proofreading; All authors have read and agreed to the published version of the manuscript.

**Funding:** This research was partially supported by the Penelitian Kolaborasi Internasional Universitas Negeri Jakarta No: 16/KI-UNJ/LPPM/IV/2021 on behalf of Dalia Sukmawati, UTM-RMC, and Arif Efektif Sdn. Bhd. with grant NOs. RJ130000.7609.4C187 and RJ130000.7344.4B200. This study was also funded by the Deanship of Scientific Research, King Saud University, through the Vice Deanship of Scientific Research Chairs.

**Institutional Review Board Statement:** Not applicable.

**Informed Consent Statement:** Not applicable.

**Data Availability Statement:** Not applicable.

**Acknowledgments:** This research was partially supported by the Penelitian Kolaborasi Internasional Universitas Negeri Jakarta No: 16/KI-UNJ/LPPM/IV/2021 on behalf of Dalia Sukmawati, UTM-RMC, and Arif Efektif Sdn. Bhd. with grant NOs. RJ130000.7609.4C187 and RJ130000.7344.4B200. This study was also funded by the Deanship of Scientific Research, King Saud University, through the Vice Deanship of Scientific Research Chairs.

**Conflicts of Interest:** The authors declare no conflict of interest.

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
