# Peer review of "Biocontrol Activity of Aureubasidium pullulans and Candida orthopsilosis Isolated from Tectona grandis L. Phylloplane against Aspergillus sp. in Post-Harvested Citrus Fruit"

_sustainability, doi:10.3390/su13137479_

Round 1
Reviewer 1 Report
Two yeasts could be used for post-harvest preservation as a natural and safe alternative compared to the currently used fungicides which have a high negative impact on the environment and human health. The results are interesting.
There are few references, especially in the last three years, If possible, please supplement the literature during 2020-2021.
Whether the management of experimental materials is consistent, and how the sample size will directly affect the experimental results, please add.
Fig.3 Disease severity value is missing SD values.
Author Response
Response to Reviewer 1 Comments
Point 1: Two yeasts could be used for post-harvest preservation as a natural and safe alternative compared to the currently used fungicides which have a high negative impact on the environment and human health. The results are interesting.
Response 1: We are thankful to the reviewer for this comment and happy to see the finding was clear to the reviewer
Point 2: There are few references, especially in the last three years, If possible, please supplement the literature during 2020-2021.
Response 2: Thank you for the reviewer and we have covered this point in the revised version by adding 5 new references (by replacing some references and adding other papers). As follows:
- Reference No. 21. Delali et al. (2021), Postharvest Biology and Technology
- Reference No. 22 Cabañas et al. (2020), Foods (MDPI journal)
- Reference No. 45. Wang et al. (2020). Food Microbiology
- Reference No. 46. Zhu et al. (2019). Biological control
- Reference No. 47 Diaz et al. (2020). Microorganisms (MDPI Journal)
Point 3: Whether the management of experimental materials is consistent, and how the sample size will directly affect the experimental results, please add.
Response 3: Thank you for this important point which we added in the materials and methods part. “Each treatment was carried out five times with 10 oranges each to ensure the reproducibility of the results”.
Point 4: Fig.3 Disease severity value is missing SD values.
Response 4. Thank you for this critical comment to give the confidence to the readers about the reproducibility of the results. Standard deviation is now added to Figure 3.
Thank you again for the great comments which help us to improve the quality of the manuscript.
Reviewer 2 Report
In this manuscript, Sukmawati et al. isolate and identify molds from Citrus sinensis, and evaluate the effectiveness of yeast obtained from T. grandis plant as biocontrol agents in vitro and in vivo. This well-written and illustrated article would be of broad interest to readers of the Sustainability journal. However, some items need to be addressed :
- It would be nice for authors to report the number of replicates in the pathogenicity test. The error bar is needed in Figure 3.
- Line 91: Authors should explain why tap water, instead of sterile water, is used for rinsing after disinfection.
- Line 101: the typo “\ycelium” needs to be correct.
- Line 249: Superscripts are needed before “CFU/mL.”
- Line 284: “Y1” should be inserted between “yeast” and “and Y10”.
- Authors have used “destructive molds” nine times and “ destructive mould” six-time. It is vital to keep them consistent in one manuscript.
Author Response
Response to Reviewer 2 Comments
In this manuscript, Sukmawati et al. isolate and identify molds from Citrus sinensis, and evaluate the effectiveness of yeast obtained from T. grandis plant as biocontrol agents in vitro and in vivo. This well-written and illustrated article would be of broad interest to readers of the Sustainability journal. However, some items need to be addressed
We are very thankful for the reviewer for this good feedback which indicates the importance and quality of work.
Point 1: It would be nice for authors to report the number of replicates in the pathogenicity test. The error bar is needed in Figure 3.
Response 1: We have considered this in the revised version by adding in Materials and Methods part: “Each treatment was carried out five times with 10 oranges each to ensure the reproducibility of the results”. We have added also the error bar for Figure 3.
Point 2: Line 91: Authors should explain why tap water, instead of sterile water, is used for rinsing after disinfection.
Response 2: This part was wrongly written in the first version. According to the protocol we used, we use distilled water. The statement now is corrected to: “Selected fruits were surface-disinfected with distilled water (5 min.); NaOCl 0.5% (1 min.); and alcohol 70% (1 min.)”.
Point 3: Line 101: the typo “\ycelium” needs to be correct.
Response 3: This type mistake has been corrected in the revised version (now line 102)
Point 4: Line 249: Superscripts are needed before “CFU/mL.”
Response 4: This typo mistake is corrected in the revised version (now line 252)
Point 5: Line 284: “Y1” should be inserted between “yeast” and “and Y10”.
Response 5: This mistake is now corrected in the revised version (now line 286)
Point 6: Authors have used “destructive molds” nine times and “destructive mould” six-time. It is vital to keep them consistent in one manuscript.
Response 6: We are very thankful for the reviewer for this critical comment to standardize the term in the manuscript. We changed now all mold to mould in the entire manuscript (text and figures).